# Joint Reasoning for Multi-Faceted Commonsense Knowledge

**Yohan Chalier**                                                    YOHAN@CHALIER.FR
**Simon Razniewski**                                         SRAZNIEW@MPI-INF.MPG.DE
**Gerhard Weikum**                                             WEIKUM@MPI-INF.MPG.DE
*Max Planck Institute for Informatics*

## Abstract

Commonsense knowledge (CSK) can support AI applications, such as image description or chatbots. Prior works on acquiring CSK, such as ConceptNet, compiled statements about concepts and their properties, but most works did not distinguish typical versus possible properties. Instead, the extraction confidence scores were used as a proxy for this kind of refinement. This paper introduces a multi-faceted model of CSK statements and methods for joint reasoning over sets of inter-related statements. Our model captures four refined facets for statements: plausibility, typicality, remarkability and salience, with scoring and ranking along each dimension. For example, hyenas drinking water is typical but not salient, whereas hyenas eating carcasses is salient. For reasoning and ranking, we develop an ILP-based method with soft constraints, to couple the inference over concepts that are related in a taxonomic hierarchy. Our evaluation shows that we can consolidate existing large CSK collections into much cleaner and more expressive knowledge.

## 1. Introduction

**Motivation and problem.** Commonsense knowledge (CSK) is a potentially important asset towards building versatile AI applications, such as visual understanding for describing images (e.g., (Agrawal et al., 2017; Karpathy and Fei-Fei, 2017; Shuster et al., 2019)) or conversational agents like chatbots (e.g., (Young et al., 2018; Rajani et al., 2019; Yang et al., 2019)). In delineation from encyclopedic knowledge on entities like Trump, Sydney, or FC Liverpool, CSK refers to properties, traits and relations of everyday concepts, such as elephants, coffee mugs or school buses. For example, when seeing scenes of an elephant juggling a few coffee mugs with its trunk, or with school kids pushing an elephant into a bus, an AI agent with CSK should realize the absurdity of these scenes and should generate adequate captions for image description or utterances in a conversation.

Encyclopedic knowledge bases (KBs) received much attention, with projects such as DBpedia, Wikidata, Yago or NELL and large knowledge graphs at Amazon, Baidu, Google, Microsoft etc. supporting entity-centric search and other services. In contrast, approaches to acquire CSK have been few and limited. Projects like ConceptNet (Speer and Havasi, 2012), WebChild (Tandon et al., 2014), TupleKB (Mishra et al., 2017) and Quasimodo (Romero et al., 2019) have compiled millions of *concept:property* or *subject-predicate-object (SPO)* statements, but still suffer from sparsity or noise. ConceptNet has only a single non-taxonomic/non-lexical statement about hyenas, namely, *hyenas: laugh a lot* [1], and WebChild

---

1. http://conceptnet.io/c/en/hyena

lists overly general and contradictory properties such as *small, large, demonic* and *fair* for hyenas[2].

Another limitation of existing CSK collections is that they rank statements solely by confidence scores. There is no information about which properties are typical, which ones are possible but rare, and which ones are salient from a human perspective. For example, the statement that hyenas drink milk (as all mammals when they are cubs) is valid, but it is not typical. Hyenas eating meat is typical, but it is not salient in the sense that humans would spontaneously name this as a key trait of hyenas. In contrast, hyenas eating carcasses is remarkable as it sets hyenas apart from other African predators (like lions or leopards), and many humans would list this as a salient property. Prior works on building Web-scale CSK collections largely missed out on this more refined perspective (see Section 2 for more discussion).

**Approach and contribution.** We present DICE (Diverse Commonsense Knowledge), a reasoning-based method for advancing CSK collections to a more expressive stage of multi-faceted knowledge. DICE is based on two novel ideas:

- We introduce four facets of concept properties: (i) *Plausibility* indicates whether a statement makes sense at all (like the established but overloaded notion of confidence scores). (ii) *Typicality* indicates whether a property holds for most instances of a concept (e.g., not only for cubs). (iii) *Remarkability* expresses that a property distinguishes the concept from closely related concepts (like siblings in a taxonomy). (iv) *salience* reflects that a property is characteristic for the concept, in the sense that most humans would list it as a key trait of the concept.
- We identify inter-related concepts by their neighborhoods in a concept hierarchy or via embeddings, and devise a set of weighted soft constraints that allows us to jointly reason over the four dimensions for sets of candidate statements. We cast this approach into an integer linear program (ILP), and harness the theory of reduced cost (Bertsimas and Tsitsiklis, 1997) for LP relaxations to compute rankings for each of the facets.

As an example, consider the concepts *lions, leopards, cheetahs* and *hyenas*. The first three are coupled by being taxonomic siblings under their hypernym *big cats*, and the last one is highly related by word-level embeddings (as all live in the African savannah). Our constraint system includes logical clauses such as

$$\text{Plausible}(s_1, p) \land \text{Related}(s_1, s_2) \land \neg\text{Plausible}(s_2, p) \land \ldots \Rightarrow \text{Remarkable}(s_1, p)$$

where $\ldots$ refers to enumerating all siblings of $s_1$, or highly related concepts. The constraint itself is weighted by the degree of relatedness; so it is a soft constraint that does allow exceptions. This way we can infer that remarkable (and also salient) statements include *lions: live in prides, leopards: climb trees, cheetahs: run fast* and *hyenas: eat carcasses*. Experiments with inputs from ConceptNet, TupleKB and Quasimodo show that DICE achieves high precision and good coverage for its multi-faceted output. The resulting datasets contain 1.6 million statements for 74K concepts, available at https://tinyurl.com/y6hygoh8.

---

2. https://gate.d5.mpi-inf.mpg.de/webchild2/?x=hyena%23n%231

## 2. Related Work

**Manually compiled CSK.** The Cyc project (Lenat, 1995) project pursued the goal of compiling a comprehensive machine-readable collection of human knowledge into logical assertions. The result comprised both encyclopedic and commonsense knowledge. The WordNet project (Miller, 1995) organized word senses into lexical relations like synonymy, antonymy, and hypernymy/hyponymy (i.e., subsumption). WordNet can serve as a taxonomic backbone for CSK, but there are also more recent alternatives such as WebIsA-LOD (Hertling and Paulheim, 2017) derived from Web contents. ConceptNet extended the Cyc and WordNet approaches by collecting CSK triples from crowdworkers, for about 20 broad predicates (Speer and Havasi, 2012). It is the state of the art for large-scale CSK. Atomic (Sap et al., 2018) is a crowdsourcing project compiling knowledge on human activities; it is more refined than ConceptNet but even sparser.

The most popular knowledge base today, Wikidata (Vrandečić and Krötzsch, 2014), contains both encyclopedic knowledge about notable entities and some CSK cast into RDF triples. However, the focus is on individual entities, and CSK is very sparse.

**Web-extracted CSK.** The reliance on human input limits the scale and scope of the above mentioned projects. Automatic extraction from Web contents can potentially achieve much higher coverage. The WebChild project (Tandon et al., 2014, 2017) extracted ca. 10 million SPO statements from books and image tags. Its rationale was to capture many plausible properties that hold for some instances of a concept, with a long tail of noisy, puzzling or invalid statements. Its approach was to identify typical statements by thresholding on confidence scores. However, these scores reflect different aspects that cannot be disentangled by mere thresholding; most critically, they are affected by the reporting bias of Web sources (Gordon and Durme, 2013). Therefore, a high-confidence statement could still be atypical (e.g., snakes attack humans). TupleKB (Mishra et al., 2017), part of the Mosaic project, contains ca. 280k statements on 8th-grade elementary science towards tackling the school exam challenge (Schoenick et al., 2017). It builds on similar sources as WebChild, but prioritizes precision over recall by supervised cleaning steps. On the downside, its coverage is much lower. Quasimodo (Romero et al., 2019) is a recent CSK collection, built by extraction from QA forums and web query logs, with ca. 4.6 million statements. It combines multiple cues into a regression-based corroboration model for ranking and aims to identify salient statements, but the model is limited by its one-dimensional scoring.

Common to these projects is that their quantitative scoring of CSK statements is based on extraction confidence only. Simply thresholding on these scores cannot bring out the refined facets of typicality and salience.

**Modalities of knowledge.** Classical works on epistemic and modal logics (see, e.g., (Fagin et al., 1995; Gabbay et al., 2003)), already proposed to add modalities for statements that hold *always* or *sometimes*. Early work on knowledge acquisition from a computational linguistics perspective (Schubert, 2002; Schubert and Tong, 2003) refined these considerations by extracting *general* versus *possible* propositions from text corpora. (Gordon and Schubert, 2010) formalized this into logical modalities like *all-or-most*, *some* etc. (Zhang et al., 2017) developed a regression model for acquiring graded modes of CSK statements, covering the spectrum of *very likely*, *likely*, *plausible*, *technically possible* and *impossible*. The model was applied to sentences for textual entailment and natural language inference

(cf. (Bowman et al., 2015)). The resulting dataset JOCI is organized in the form of 40K sentence pairs, rather than grouped by concepts and systematically covering their properties.

All these efforts were relatively small-scale by modern standards. Our approach follows up on this inspiring line of works but extends their scope and scale, by including human-perceived salience, by leveraging concept taxonomies to enhance coverage and accuracy, and by devising a methodology that can process millions of CSK statements.

## 3. Multi-Faceted CSK Model

We consider simple CSK statements of the form $(s, p)$, where $s$ is a concept and $p$ is a property of this concept. We refer to $s$ as the subject of the statement. Often, but not necessarily, $s$ is a single noun, such as *hyenas*, but multi-word phrases do occur, such as *African elephants*. $p$ values include many verb or noun phrases, such as *laugh a lot* or *(are) endangered species*.

Unlike prior works, we do not adopt the SPO triple model, and instead combine P and O into *property*, for three reasons:
(i) The split between P and O is often arbitrary. For *lions : live in prides*, we could either consider *live* or *live in* as P and the rest as O, or we view *live in prides* as P without O.
(ii) Unlike encyclopedic KBs, CSK is so diverse that it is virtually impossible to agree on canonicalized predicate names beyond a small set of basic properties. ConceptNet and WebChild cover a fixed set of ca. 20 pre-specified predicates. When discounting taxonomic and lexical relations (e.g., type-of, synonymy), this boils down to a few broad predicates: *used for*, *capable of*, *location*, *part of* and the generic *has property*.
(iii) In contrast, TupleKB and Quasimodo have more than 1000 distinct predicates. The simpler model of concept:property pairs makes all these CSK collections better comparable by unifying their highly varying granularities.

### 3.1 CSK Dimensions

We organize concept-property pairs along four dimensions: plausibility (Tandon et al., 2014; Mishra et al., 2017), typicality (Speer and Havasi, 2012; Zhang et al., 2017), salience (Romero et al., 2019), and remarkability (information theory). These are meta-properties: each $(s, p)$ pair can have any of these labels and multiple labels are possible. The first three modalities together cover the foci of most prior works on large CSK collections, whereas each of these works emphasizes only one or two of the modalities. Remarkability is a new dimension, motivated by our novel way of leveraging class taxonomies.

For each statement and dimension label, we compute a score and can thus rank statements for a concept four times.

- *Plausibility:* Is the property valid at least for some instances of the concept, for at least some spatial, temporal or socio-cultural contexts? For example, lions drink milk at some time in their lives, and some lions attack humans.
- *Typicality:* Does the property hold for most (or ideally all) instances of the concept, for most contexts? For example, most lions eat meat, regardless of whether they live in Africa or in a zoo.

- *Remarkability:* What are specific properties of a concept that sets the concept apart from highly related concepts, like taxonomic generalizations (hypernyms in a concept hierarchy)? For example, lions live in prides but not other big cats do this, and hyenas eat carcasses but hardly any other African predator does this.
- *Salience:* When humans are asked about a concept, such as *lions*, *bicycles* or *rap songs*, would a property be listed among the concept's most notable traits, by most people? For example, lions hunt in packs, bicycles have two wheels, rap songs have interesting lyrics and beat (but no real melody).

**Examples.** Refining CSK by the four dimensions is useful for various application areas, including language understanding for chatbots, as illustrated by the following examples:

- Plausibility helps to detect absurd statements and react adequately.
- Typicality helps a chatbot to infer missing context. For example, when the human talks about "a documentary which showed the feeding frenzy of a pack of hyenas", the chatbot could ask "what kind of carcass did they feed on?"
- Remarkability can be a signal for the chatbot to infer which concept the human is talking about. For example, a user utterance "In the zoo, the kids where fascinated by a spotted dog that was laughing at them" could lead to chatbot to detect that this is about hyenas.
- Salience enables the chatbot to focus on key properties. For example, when talking about lions in the zoo, the bot could ask "Did you hear the lion roar?", or "How many lionesses were in the king's harem?"

## 4. Joint Reasoning

**Overview.** For reasoning over sets of CSK statements, we start with a CSK collection, like ConceptNet, TupleKB or Quasimodo. These are in triple form with crisp subjects but potentially noisy phrases as predicates and objects. We interpret each subject as a concept and concatenate the predicate and object into a property. Inter-related subsets of statements are identified by locating concepts in a large taxonomy and grouping siblings and their hypernymy parents together. These groups may overlap. For this purpose we use the WebIsALOD taxonomy (Hertling and Paulheim, 2017), as it has very good coverage of concepts and captures everyday vocabulary.

Based on the taxonomy, we also generate additional candidate statements for sub- or super-concepts, as we assume that many properties are inherited between parent and child. We use rule-based templates for this expansion of the CSK collection (e.g., as lions are predators, big cats and also tigers, leopards etc. are predators as well). This mitigates the sparseness in the observation space. Note that, without the reasoning, this would be a high-risk step as it includes many invalid statements (e.g., lions live in prides, but big cats in general do not). Reasoning will prune out most of the invalid candidates, though.

For joint reasoning over the statements for the concepts of a group, we interpret the rule-based templates as soft constraints, with appropriate weights.

For setting weights in a meaningful way, we leverage prior scores that the initial CSK statements come with (e.g., confidence scores from ConceptNet), and additional statistics from large corpora, most notably word-level embeddings like word2vec.

In this section, we develop the logical representation and the joint reasoning method, assuming that we have weights for statements and for the grounded instantiations of the constraints. Subsequently, Section 5 presents techniques for obtaining statistical priors for setting the weights.

### 4.1 Coupling of CSK Dimensions

Let $\mathcal{S}$ denote the set of subjects and $\mathcal{P}$ the properties. The inter-dependencies between the four CSK dimensions are expressed by the following logical constraints.

**Concept-dimension dependencies:** $\forall (s, p) \in \mathcal{S} \times \mathcal{P}$

$$\text{Typical}(s, p) \Rightarrow \text{Plausible}(s, p) \tag{1}$$

$$\text{Salient}(s, p) \Rightarrow \text{Plausible}(s, p) \tag{2}$$

$$\text{Typical}(s, p) \wedge \text{Remarkable}(s, p) \Rightarrow \text{Salient}(s, p) \tag{3}$$

These clauses capture the intuition behind the four facets.

**Parent-child dependencies:** $\forall (s_1, p) \in \mathcal{S} \times \mathcal{P}, \forall s_2 \in \text{children}(s_1)$

$$\text{Plausible}(s_1, p) \Rightarrow \text{Plausible}(s_2, p) \tag{4}$$

$$\text{Typical}(s_1, p) \Rightarrow \text{Typical}(s_2, p) \tag{5}$$

$$\text{Typical}(s_2, p) \Rightarrow \text{Plausible}(s_1, p) \tag{6}$$

$$\text{Remarkable}(s_1, p) \Rightarrow \neg\text{Remarkable}(s_2, p) \tag{7}$$

$$\text{Typical}(s_1, p) \Rightarrow \neg\text{Remarkable}(s_2, p) \tag{8}$$

$$\neg\text{Plausible}(s_1, p) \wedge \text{Plausible}(s_2, p) \Rightarrow \text{Remarkable}(s_2, p) \tag{9}$$

$$(\forall s_2 \in \text{children}(s_1) \ \text{Typical}(s_2, p)) \Rightarrow \text{Typical}(s_1, p) \tag{10}$$

These dependencies state how properties are inherited between a parent concept and its children in a taxonomic hierarchy. For example, if a property is typical for the parent and thus for all its children, it is not remarkable for any child as it does not set any child apart from its siblings.

**Sibling dependencies:** $\forall (s_1, p) \in \mathcal{S} \times \mathcal{P}, \forall s_2 \in \text{siblings}(s_1)$

$$\text{Remarkable}(s_1, p) \Rightarrow \neg\text{Remarkable}(s_2, p) \tag{11}$$

$$\text{Typical}(s_1, p) \Rightarrow \neg\text{Remarkable}(s_2, p) \tag{12}$$

$$\neg\text{Plausible}(s_1, p) \wedge \text{Plausible}(s_2, p) \Rightarrow \text{Remarkable}(s_2, p) \tag{13}$$

These dependencies state how properties of concepts under the same parent relate to each other. For example, a property being plausible for only one in a set of siblings makes this property remarkable for the one concept.

### 4.2 Grounding of Dependencies

The specified first-order constraints need to be grounded with the candidate statements in a CSK collection, yielding a set of logical clauses (i.e., disjunctions of positive or negated atomic statements). To avoid producing a huge amount of clauses, we restrict the grounding to existing subject-property pairs and the high-confidence ($>0.4$) relationships of the WebIsALOD taxonomy (avoiding its noisy long tail).

| Rule | Clause | $\omega_r$ | $\omega_s$ | $\omega_e$ | $\omega^c$ |
|------|--------|-----------|-----------|-----------|-----------|
| 1 | Plausible(*car, hit wall*) $\vee \neg$ Typical(*car, hit wall*) | 0.48 | 1 | 0.60 | 0.29 |
| 14a | Plausible(*bicycle, be at city*) $\vee \neg$ Plausible(*bicycle, be at town*) | 0.85 | 0.86 | 1 | 0.73 |
| 14a | Plausible(*bicycle, be at town*) $\vee \neg$ Plausible(*bicycle, be at city*) | 0.85 | 0.86 | 1 | 0.73 |
| 8 | $\neg$ Remarkable (*bicycle, transport person and thing*) $\vee \neg$ Typical(*car, move person*) | 0.51 | 0.78 | 0.96 | 0.38 |

Table 1: Examples of grounded clauses with their weights (based on ConceptNet).

**Expansion to similar properties.** Following this specification, the clauses would apply only for the same property of inter-related concepts, for example, *eats meat* for *lions, leopards, hyenas* etc. However, the CSK candidates may express the same or very similar properties in different ways: *lions: eat meat, leopards: are carnivores, hyenas: eat carcasses* etc. Then the grounded formulas would never trigger any inference, as the $p$ values are different. We solve this issue by considering the similarity of different $p$ values based on word-level embeddings (see Section 5). For each property pair $(p_1, p_2) \in \mathcal{P}^2$, grounded clauses are generated if $\text{sim}(p_1, p_2)$ exceeds a threshold $t$.

We consider such highly related property pairs also for each concept alone, so that we can deduce additional CSK statements by generating the following clauses: $\forall s \in \mathcal{S}, \forall (p, q) \in \mathcal{P}^2$,

$$\text{sim}(p, q) \geq t \Rightarrow (\text{Plausible}(s, p) \Leftrightarrow \text{Plausible}(s, q)), \tag{14a}$$

$$(\text{Typical}(s, p) \Leftrightarrow \text{Typical}(s, q)), \tag{14b}$$

$$(\text{Remarkable}(s, p) \Leftrightarrow \text{Remarkable}(s, q)), \tag{14c}$$

$$(\text{Salient}(s, p) \Leftrightarrow \text{Salient}(s, q)) \tag{14d}$$

This expansion of the reasoning machinery allows us to deal with the noise and sparsity in the pre-existing CSK collections.

**Weighting clauses.** Each of the statements Plausible$(s, p)$, Typical$(s, p)$, Remarkable$(s, p)$ and Salient$(s, p)$ has a prior weight based on the confidence score from the underlying CSK collection (see Sec. 5). These priors are denoted $\pi(s, p), \tau(s, p), \rho(s, p)$, and $\sigma(s, p)$.

Each grounded clause $c$ has three different weights: (i) $\omega_r$, the weight of the logical dependency from which the clause is generated, a hyper-parameter for tuning the relative influence of different kinds of dependencies. (ii) $\omega_s$, the similarity weight, $\text{sim}(p_1, p_2)$ for clauses resulting from similarity expansion, or 1.0 if concerning only a single property. (iii) $\omega_e$, the evidence weight, computed by combining the statistical priors for the individual atoms of the clause, using basic probability calculations for logical operators: $1 - u$ for negation and $u + v - uv$ for disjunction with weights $u, v$ for the atoms in a clause. The final weight of a clause $c$ is computed as $\omega^c = \omega_r \omega_s \omega_e$. Table 1 shows a few illustrative examples.

## 4.3 ILP encoding and solving

The resulting maxSAT problems are encoded into an ILP (Vazirani, 2013) (Integer Linear Program), which in turn is relaxed into a linear program, thus allowing to exploit the theory of reduced costs for ranking (Bertsimas and Tsitsiklis, 1997). As the resulting programs are typically intractable, we partitioned the problem into multiple sub-problems, each representing one concept along with its close taxonomic neighbourhood. The sub-problems were then solved using the Gurobi Optimizer. Further details on the implementation are in Appendix A.

We also considered using other kinds of probabilistic-logical reasoning, such as PSL (Bach et al., 2017), but our objective requires more than MAP inference and so would still face tractability issues. Moreover, with the input partitioning sketched above, the industrial-strength ILP solver gave us sufficient scalability.

## 5. Prior Statistics

So far, we assumed that prior scores – $\pi(s,p), \tau(s,p), \rho(s,p), \sigma(s,p)$ – are given, in order to compute weights for the ILP or LP. This section explains how we obtain these priors. In a nutshell, we obtain basic scores from the underlying CSK collections and their combination with embedding-based similarity, and from textual entailment and relatedness in the taxonomy (Subsection 5.1). We then define aggregation functions to combine these various cues (Subsection 5.2).

### 5.1 Basic Scores

Basic statements like $(s,p)$ are taken from existing CSK collections, which often provide *confidence scores* based on observation frequencies or human assessment (of crowdsourced statements or samples). We combine these confidence measures, denoted $\text{score}(s,p)$ with embedding-based similarity between two properties, $\text{sim}(p,q)$. Each property $p$ is tokenized into a bag-of-words $\{w_1, \ldots, w_n\}$ and encoded as the idf-weighted centroid of the embedding vectors $\vec{w}_i$ obtained from a pretrained GoogleNews word2vec model: $\vec{p} = \sum_{i=1}^{n} \text{idf}(w_i)\,\vec{w}_i$. The similarity between two properties is the cosine between the vectors mapped into $[0,1]$: $\text{sim}(p,q) = \frac{1}{2}\left(\frac{\langle \vec{p}, \vec{q} \rangle}{\|\vec{p}\|\|\vec{q}\|} + 1\right)$.

Confidence scores and similarities are combined into a quasi-probability:

$$\mathbb{P}[s,p] = \frac{1}{Z} \sum_{q \in \mathcal{P},\ \text{sim}(p,q) \geq t} \text{score}(s,q) \times \text{sim}(q,p)$$

where $Z$ is a normalization factor and $t$ is a threshold (set to 0.75 in our implementation). The intuition for this measure is that it reflects the probability of $(s,p)$ being observed in the digital world, where evidence is accumulated over different phrases for inter-related properties such as *eat meat*, *are carnivores*, *are predators*, *prey on antelopes* etc.

We can now derive additional measures that serve as building blocks for the final priors:
- the marginals $\mathbb{P}[s]$ for subjects and $\mathbb{P}[p]$ for properties,
- the conditional probabilities of observing $p$ given $s$, or the reverse; $\mathbb{P}[p \mid s]$ can be thought of as the *necessity* of the property $p$ for the subject $s$, while $\mathbb{P}[s \mid p]$ can be thought of as a *sufficiency* measure,
- the probability that the observation of $s$ implies the observation of $p$, which can be expressed as $\mathbb{P}[s \Rightarrow p] = 1 - \mathbb{P}[s] + \mathbb{P}[s,p]$.

Beyond aggregated frequency scores, priors rely on two more components, scores from textual entailment models and taxonomy-based information gain.

**Textual entailment:**
A variant of $P[s \Rightarrow p]$ is to tap into corpora and learned models for textual entailment: does a sentence such as "Simba is a lion" entail a sentence "Simba lives in a pride"? We

| Dimension | Random | ConceptNet | | TupleKB | | Quasimodo | | Music-manual | |
|---|---|---|---|---|---|---|---|---|---|
| | | Baseline | Dice | Baseline | Dice | Baseline | Dice | Baseline | Dice |
| Plausible | 0.5 | 0.52 | 0.62 | 0.53 | 0.57 | 0.57 | 0.59 | 0.21 | **0.67** |
| Typical | 0.5 | 0.39 | **0.65** | 0.37 | 0.59 | 0.52 | **0.64** | 0.54 | **0.70** |
| Remarkable | 0.5 | 0.52 | **0.69** | 0.50 | 0.54 | 0.56 | 0.56 | 0.49 | **0.74** |
| Salient | 0.5 | 0.54 | 0.65 | 0.59 | 0.61 | 0.53 | 0.63 | 0.51 | 0.65 |
| Avg. | 0.5 | 0.50 | **0.66** | 0.50 | 0.58 | 0.54 | 0.61 | 0.52 | **0.69** |

Table 2: Precision of pairwise preference (ppref) of DICE versus original CSK collections. Statistically significant gains over baselines (by one-tailed t-test level $\alpha = 0.05$) are in boldface.

leverage the attention model from the AllenNLP project (Gardner et al., 2018) learned from the SNLI corpus (Bowman et al., 2015) and other annotated text collections. This gives us scores for two measures: does $s$ entail $p$, entail$(s \to p)$, and does $p$ contradict $s$, con$(s, p)$.

**Taxonomy-based information gain:**
For each $(s, p)$ we define a neighborhood of concepts, $N(s)$, by the parents and siblings of $s$, and consider all statements for $s$ versus all statements for $N(s) - \{s\}$ as a potential cue for remarkability. For each property $p$ and concept set $S$, the entropy of $p$ is $H(p|S) = \frac{1}{X_S} \log X_S + \frac{X_S - 1}{X_S} \log \frac{X_S}{X_S - 1}$ where $X_S = |\{q \mid \exists s \in S : (s, q)\}|$. Instead of merely count-based entropy, we could also incorporate relative weights of different properties, but the as a basic cue, the simple measure is sufficient. Then, the information gain of $(s, p)$ is $IG(s, p) = H(p \mid \{s\}) - H(p \mid S - \{s\})$.

## 5.2 Score Aggregation

All the basic scores – $\mathbb{P}[s, p]$, $\mathbb{P}[s \mid p]$, $\mathbb{P}[p \mid s]$, $\mathbb{P}[s \Rightarrow p]$, entail$(s \to p)$, con$(s, p)$ and $IG(s, p)$ – are fed into regression models that learn an aggregate score for each of the four facets: plausibility, typicality, remarkability and salience. The regression parameters (i.e., weights for the different basic scores) are learned from small set of facet-annotated CSK statements, separately, for each of the four facets. We denote the aggregated scores, serving as priors for the reasoning step, as $\pi(s, p)$, $\tau(s, p)$, $\rho(s, p)$ and $\sigma(s, p)$.

## 6. Experiments

We evaluated how well DICE ranks statements in four CSK collections along the four facets. We sampled 200 subjects each with two properties per CSK collection, and asked 3 annotators for their preference among the two properties on each dimension (e.g., "Which one is more salient: *lions: hunt zebras* or *lions: sleep at night*"). We evaluated the precision in pairwise preference (ppref) of DICE scores, compared to the original one-dimensional scores of the input CSK collections. The results are shown in Table 2. DICE outperforms the original rankings by a wide margin, and performs best when given a clean taxonomy as input (Music-manual, the other datasets being based on automatically computed WebIsALOD taxonomies). Details on experiments are in Appendix B.

## 7. Discussion

**Experimental results.** The experiments showed that DICE can capture CSK along the four dimensions significantly better than the single-dimensional baselines. An ablation study highlighted that a combination of prior scoring and constraint-based joint reasoning is highly beneficial (0.66 average ppref vs. 0.58 and 0.51 of each step in isolation, see Table 5). We also performed an ablation study using only priors and only reasoning: both of these variants performed substantially worse, losing 8 and 15 percentage points, respectively, compared to the full-fledged method, with one-tailed t-test level $\alpha=0.01$.

Among the dimensions, we found that plausibility was the most difficult of the four dimensions (see Table 2). The learning of hyper-parameters showed that all constraints are useful and contribute to the outcome of DICE, with similarity dependencies and plausibility inference having the strongest influence. Comparing the three CSK collections that we worked with, we observe that the crowdsourced ConceptNet is a priori cleaner and hence easier to process than Quasimodo and TupleKB. Also, manually designed taxonomies gave DICE a performance bost of 0.03-0.11 in ppref over the noisy WebIsALOD taxonomies.

**Task difficulty.** Scoring commonsense statements by dimensions beyond confidence has never been attempted before, and a major challenge is to design appropriate and varied input signals towards specific dimensions. Our experiments showed that DICE can approximate the human-generated ground-truth rankings to a considerable degree (0.58-0.69 average ppref), although a gap remains (see Table 2). We conjecture that in order to approximate human judgments even better, more and finer-grained input signals, for example about textual contexts of statements, are needed.

**Enriched CSK data and demo.** Along with this paper, we publish six datasets: the 3 CSK collections ConceptNet, TupleKB and Quasimodo enriched by DICE with scores for the four CSK dimensions, and additional inferred statements that expand the original CSK data by about 50% (see Appendix B.2). The datasets can be downloaded from https://tinyurl.com/y6hygoh8. Code and a web-based UI will be made available (see Appendix C).

## 8. Conclusion

This paper presented DICE, a joint reasoning framework for commonsense knowledge (CSK) that incorporates inter-dependencies between statements by taxonomic relatedness and other cues. This way we can capture more expressive meta-properties of concept-property statements along the four dimensions of plausibility, typicality, remarkability and salience. This richer knowledge representation is a major advantage over prior works on CSK collections. In addition, we have devised techniques to compute informative rankings for all four dimensions, using the theory of reduced costs for LP relaxation. We believe that such multi-faceted rankings of CSK statements are crucial for next-generation AI, particularly towards more versatile and robust conversational bots. Our future work plans include leveraging this rich CSK for advanced question answering and human-machine dialogs.

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

## Appendix A. Implementation

### A.1 Integer Linear Program

**Notations.** For reasoning over the validity of candidate statements, for each of the four facets, we view every candidate statement $Facet(s, p)$ as a variable $v \in \mathcal{V}$, and its prior (either $\tau$, $\pi$, $\rho$ or $\sigma$, see Section 5) is denoted as $\omega^v$. Every grounded clause $c \in \mathcal{C}$, normalized into a disjunctive formula, can be split into variables with positive polarity, $c^+$, and variables with negative polarity, $c^-$.

By viewing all $v$ as Boolean variables, we can now interpret the reasoning task as a weighted maximum satisfiability (Max-Sat) problem: find a truth-value assignment to the variables $v \in \mathcal{V}$ such that the sum of weights of satisfied clauses is maximized. This is a classical NP-hard problem, but the literature offers a wealth of approximation algorithms (see, e.g., (Manquinho et al., 2009)). Alternatively and preferably for our approach, we can re-cast the Max-Sat problem into a problem for integer linear programming (ILP) (Vazirani, 2013) where the variables $v$ become 0-1 decision variables. Although ILP is more general and potentially more expensive than Max-Sat, there are highly optimized and excellently engineered methods available in software libraries like Gurobi (Gurobi Optimization, 2019). Moreover, we are ultimately interested not just in computing accepted variables (set to 1) versus rejected ones (set to 0), but want to obtain an informative ranking of the candidate statements. To this end, we can relax an ILP into a fractional LP (linear program), based on principled foundations (Vazirani, 2013), as discussed below. Therefore, we adopt an ILP approach, with the following objective function and constraints:

$$\max \sum_{v \in \mathcal{V}} \omega^v v + \sum_{c \in \mathcal{C}} \omega^c c \tag{15}$$

under the constraints:

$$\forall c \in \mathcal{C} \ \ \forall v \in c^+ \ \ c - v \geq 0 \tag{16a}$$

$$\forall c \in \mathcal{C} \ \ \forall w \in c^- \ \ c + w - 1 \geq 0 \tag{16b}$$

$$\forall c \in \mathcal{C} \ \ \sum_{v \in c^+} v + \sum_{w \in c^-} (1 - w) - c \geq 0 \tag{16c}$$

$$\forall v \in \mathcal{V} \ \ v \in [0, 1] \tag{16d}$$

$$\forall c \in \mathcal{C} \ \ c \in [0, 1] \tag{16e}$$

Each clause $c$ is represented as a triple of ILP constraints, where Boolean operations $\neg$ and $\vee$ are encoded via inequalities.

### A.2 Ranking of CSK Statements

The ILP returns 0-1 values for the decision variables; so we can only accept or reject a candidate statement. Relaxing the ILP into an ordinary linear program (LP) drops the integrality constraints on the decision variables, and would then return fractional values for the variables. Solving an LP is typically faster than solving an ILP.

The fractional values returned by the LP are not easily interpretable. We could employ the method of randomized rounding (Raghavan and Thompson, 1987): for fractional value

$x \in [0,1]$ we toss a coin that shows 1 with probability $x$ and 0 with probability $1-x$. This has been proven to be a constant-factor approximation (i.e., near-optimal solution) on expectation.

However, we are actually interested in using the relaxed LP to compute principled and informative rankings for the candidate statements. To this end, we leverage the theory of *reduced costs*, aka. *opportunity costs* (Bertsimas and Tsitsiklis, 1997). For an LP of the form *minimize $c^T x$ subject to $Ax \leq b$ and $x \geq b$* with coefficient vectors $c, b$ and coefficient matrix $A$, the reduced cost of variable $x_i$ that is zero in the optimal solution is the amount by which the coefficient $c_i$ needs to be reduced in order to yield an optimal solution with $x_i > 0$. This can be computed for all $x$ as $c - A^T y$. For maximization problems, the reduced cost is an increase of $c$. Modern optimization tools like Gurobi directly yield these measures of sensitivity as part of their LP solving.

We use the reduced costs of the $x_i$ variables as a principled way of ranking them; lowest cost ranking highest (as their weights would have to be changed most to make them positive in the optimal solution).
As all variables with reduced cost zero would have the same rank, we use the actual variable values (as a cue for the corresponding statement or dependency being satisfied) as a tie-breaker.

### A.3 Scalability

LP solvers are not straightforward to scale to cope with large amounts of input data. For reasoning over all candidate statements in one shot, we would have to solve an LP with millions of variables. We devised and utilized the following technique to overcome this bottleneck in our experiments.

The key idea is to consider only limited-size neighborhoods in the taxonomic hierarchy in order to partition the input data. In our implementation, to reason about the facets for a candidate statement $(s, p)$, we identify the parents and siblings of $s$ in the taxonomy and then compile all candidate statements and grounded clauses where at least one of these concepts appears. This typically yields subsets of size in the hundreds or few thousands. Each of these forms a partition, and we generate and solve an LP for each partition separately. This way, we can run the LP solver on many partitions independently in parallel. The partitions overlap in terms of interrelated statements, but each $(s, p)$ is associated with a primary partition with this statement's specific neighborhood.

## Appendix B. Details on Experiments

We evaluate three aspects of the Dice framework: (i) accuracy in ranking statements along the four CSK facets, (ii) run-time and scalability, (iii) the ability to enrich CSK collections with newly inferred statements. The main hypothesis under test is how well Dice can rank statements for each of the four CSK facets. We evaluate this by obtaining crowdsourced judgements for a pool of sample statements.

| CSK collection | #subjects | #statements |
|---|---|---|
| Quasimodo | 13,387 | 1,219,526 |
| ConceptNet | 45,603 | 223,013 |
| TupleKB | 28,078 | 282,594 |

Table 3: Input CSK collections.

| CSK collection | #nodes | #parents/node | #siblings/node |
|---|---|---|---|
| Quasimodo | 11148 | 15.33 | 3627.8 |
| ConceptNet | 41451 | 1.15 | 63.7 |
| TupleKB | 26100 | 2.14 | 105.1 |
| Music-manual | 8 | 1.68 | 3.4 |

Table 4: Taxonomy statistics.

## B.1 Setup

**Datasets.** We use three CSK collections for evaluating the added value that DICE provides: (i) ConceptNet, a crowdsourced, sometimes wordy collection of general-world CSK. (ii) Tuple-KB, a CSK collection extracted from web sources with focus on the science domain, with comparably short and canonicalized SPO triples. (iii) Quasimodo, a web-extracted general-world CSK collection with focus on salience. Statistics on these datasets are shown in Table 3.

To construct taxonomies for each of these collections, we utilized the WebIsALOD dataset (Hertling and Paulheim, 2017), a web-extracted noisy set of ranked subsumption pairs (e.g., `tiger isA big_cat` - 0.88, `tiger isA carnivore` - 0.83). We prune out long-tail noise by setting a threshold of 0.4 for the confidence scores that WebIsALOD comes with. To evaluate the influence of taxonomy quality, we also hand-crafted a small high-quality taxonomy for the music domain, with 10 concepts and 9 subsumption pairs, such as `rapper` being a subclass of `singer`. Table 4 gives statistics on the taxonomies per CSK collection. Differences between #nodes in Table 4 and #subjects in Table 3 are caused by merging nodes on hypernymy paths without branches (#children=1).

**Annotation.** To obtain labelled data for hyper-parameter tuning and as ground-truth for evaluation, we conducted a crowdsourcing project using Amazon Mechanical Turk (AMT). For salience, typicality and remarkability, we sampled 200 subjects each with 2 properties from each of the CSK collections, and asked annotators for pairwise preference with regard to each of the three facets, using a 5-point Likert scale. That is, we show two statements for the same subject, and the annotator could slide on the scale between 1 and 5 to indicate the more salient/typical/remarkable statement. For the plausibility dimension, we sampled 200 subjects each with two properties, and asked annotators to assess the plausibility of individual statements on a 5-point scale. Then we paired up two statements for the same subject as a post-hoc preference pair. The rationale for this procedure is to avoid biasing the annotator in judging plausibility by showing two statements at once, whereas it is natural to compare pairs on the other three dimensions.

In total, we had $4 \times 4 \times 200 = 3200$ tasks, each given to 3 annotators. The final scores for each statement and facet were the averages of the three numerical judgments. Regarding inter-annotator agreement, we observed a reasonably low standard deviation of

0.81/0.92/0.98/0.92 (over the scale from 1 to 5) for the dimensions plausibility/typicality/-remarkability/salience on ConceptNet, with similar values on the other CSK collections. When removing indeterminate samples, with avg. score between 2.5 and 3.5, and interpreting annotator scores as binary preferences, inter-annotator agreement was fair to moderate, with Fleiss' Kappa values of 0.31, 0.30, 0.25 and 0.48 for plausibility, typicality, remarkability and salience, respectively.

**Guidelines for annotators.** The AMT workers were given informal explanations of the facets to be assessed, along with clear examples and justifications for each facet:

- For typicality, *birds can fly* is more typical than *birds eat bread*, because only birds in contact with humans (e.g., in city parks) can eat bread, but almost all birds can fly.
- For salience, *birds have feathers* is more salient than *birds have bones*, because the former is way more characteristic and useful in describing a bird by the color, length and shape of its feathers.
- For remarkability, *birds whistle in the forest* is more remarkable than *birds lay eggs*, because other animals like fish and reptiles lay eggs as well, whereas whistling in the forest is unique for birds.

**Evaluation Metrics.** In the actual evaluation, we used withheld pairwise annotations for statements along the dimensions plausibility, typicality, remarkability and salience as ground truth, and compared, for each system score, for how many of these pairs its scores implicated the same ordering, i.e., measured the *precision in pairwise preference* (ppref) (Carterette et al., 2008).

**Hyper-parameter tuning.** The 800 labeled statements per CSK collection were split into 70% for hyper-parameter optimization and 30% for evaluation. We performed two hyper-parameter optimization steps. In step 1, we learned the weights for aggregating the basic scores by a regression model based on interpreting pairwise data as single labels (i.e., the preferred property is labelled as 1, the other one as 0). In step 2, we used Bayesian optimization to tune the weights of the constraints. As exhaustive search was not possible, we used the Tree-structured Parzen Estimator (TPE) algorithm from the Hyperopt (Bergstra et al., 2013) library. We used the 0-1 loss function on the ordering of the pairs as metric, and explored the search space in two ways: (i) discrete exploration space $\{0, 0.1, 0.5, 1\}$, followed by (ii) continuous exploration space of radius 0.2 centered on the value selected in the previous step. For ConceptNet, constraints were assigned an average weight of 0.404, with the highest weights for: (14) Similarity constraints (weight 0.85), (6) Plausibility inference (weight 0.66) and (13) Sibling implausibility implying remarkability (weight 0.60). All constraints were assigned non-negligible positive weights; so they are all important for joint inference.

## B.2 Results

**Quality of rankings.** Table 2 shows the main result of our experiments: the precision in pairwise preference (ppref) scores (Carterette et al., 2008), that is, the fraction of pairs where DICE or a baseline produced the same ordering as the crowdsourced ground-truth. As baseline, we rank all statements by the confidence scores from the original CSK collections, which implies that the ranking is identical for all four dimensions. As the table shows,

|  | Priors only | Constraints only | Both |
|---|---|---|---|
| Plausible | 0.54 | 0.51 | 0.62 |
| Typical | 0.53 | 0.42 | 0.65 |
| Remarkable | 0.65 | 0.57 | 0.69 |
| Salient | 0.56 | 0.52 | 0.65 |
| Avg. | 0.58 | 0.51 | 0.66 |

Table 5: Ablation study using ConceptNet as input.

| Ranking dimension | Existing statements | New statements 25% | New statements 50% | New statements 100% |
|---|---|---|---|---|
| Plausible | 3.44 | 3.54 | 3.43 | 3.41 |
| Typical | 3.44 | 3.27 | 3.31 | 3.26 |

Table 6: Plausibility (Likert-scale values) of top-ranked newly inferred statements with ConceptNet as input.

| Subject | Novel properties |
|---|---|
| sculpture | be at art museum, be silver or gold in color |
| athlete | requires be good sport, be happy when they win |
| saddle | be used to ride horse, be set on table |

Table 7: Examples of new statements inferred by DICE with ConceptNet as input.

| Subject | Property | Baseline CN-score | Dice plausible | Dice typical | Dice remarkable | Dice salient |
|---|---|---|---|---|---|---|
| snake | be at shed | 0.46 | 0.29 | 0.71 | 0.29 | 0.18 |
| snake | be at pet zoo | 0.46 | 0.15 | 0.29 | 0.82 | 0.48 |
| snake | bite | 0.92 | 0.58 | 0.13 | 0.61 | 0.72 |
| lawyer | study legal precedent | 0.46 | 0.25 | 0.73 | 0.37 | 0.18 |
| lawyer | prove that person be guilty | 0.46 | 0.06 | 0.47 | 0.65 | 0.40 |
| lawyer | present case | 0.46 | 0.69 | 0.06 | 0.79 | 0.75 |
| bicycle | requires coordination | 0.67 | 0.62 | 0.40 | 0.36 | 0.35 |
| bicycle | be used to travel quite long distance | 0.46 | 0.30 | 0.20 | 0.77 | 0.64 |
| bicycle | be power by person | 0.67 | 0.19 | 0.33 | 0.66 | 0.55 |

Table 8: Anecdotal examples from DICE run on ConceptNet.

DICE consistently outperforms the baselines by a large margin of 7 to 18 percentage points. In Table 2 all numbers in boldface are statistically significant by one-tailed t-test level $\alpha = 0.05$. It is also notable that scores in the original ConceptNet and TupleKB are negatively correlated with typicality (values lower than 0.5), pointing out a substantial fraction of valid but not exactly typical properties in these pre-existing CSK collections.

**Ablation study.** To study the impact of statistical priors and constraint-based reasoning, we compare two variants of DICE: (i) using only priors without the reasoning stage, and (ii) using only the constraint-based reasoning with all priors set to 0.5. The resulting ppref scores are shown in Table 5. In isolation, priors and reasoning perform 8 and 15 percentage points worse than the combined DICE method. These differences are statistically significant for one-tailed t-test level $\alpha=0.01$. This clearly demonstrates the importance of both stages and the synergistic benefit from their interplay.

**Enrichment potential.** All CSK collections are limited in their coverage of long-tail concepts. By exploiting the taxonomic and embedding-based relatedness between different concepts, we can generate candidate statements that were not observed before (e.g., because online contents rarely talk about generalized concepts like big cats, and mostly mention only properties of lions, leopards, tigers etc.). As mentioned in Section 4.2, templates can be used to generate candidates. We apply templates for all parents, grand-parents, siblings and children of a concept to generate candidate statements. These noisy candidates are then fed into DICE reasoning together with the statements that are actually contained in the existing CSK collections. This yields cleaner subsets of new statements.

To evaluate the quality of the DICE output for such "unobserved" statements, we randomly sampled 10 ConceptNet subjects, and grounded the reasoning framework for these subjects for all properties observed in their taxonomic neighbourhood (i.e., parents and siblings). We then asked annotators to assess the plausibility of 100 sampled statements.

To compute the quality of DICE scores, we consider the top-ranked statements by predicted plausibility and by typicality, where we vary the recall level: number of statements from the ranking in relation to the number of statements that ConceptNet contains for the sampled subjects. The Likert-scale values from AMT-based assessment are shown in Table 6 for recall 25%, 50% and 100%, that is up to doubling the size of ConceptNet for the given subjects. As one can see, DICE can expand the pre-existing CSK by 25% without losing in quality, and even up to 100% expansion the decrease in quality is negligible. Table 7 presents anecdotal statements absent in ConceptNet.

**Run-Time.** All experiments were run on a cluster with 40 cores and 500 GB memory. Hyper-parameter optimization took 10-14 hours for each of the three CSK inputs. Computing the four-dimensional scores for all statements took about 3 hours, 3 hours and 24 hours for ConceptNet, TupleKB and Quasimodo, respectively.

The computationally most expensive steps are the semantic similarity computation and the LP solving. For semantic similarity computation, a big handicap is the verbosity and hence diversity of the phrases for properties (e.g., "live in the savannah", "roam in the savannah", "are seen in the African savannah", "can be found in Africa's grasslands" etc.). We observed on average 1.55 statements per distinct property for ConceptNet, and 1.77 for Quasimodo. Therefore, building the input matrix for the LP is very time-consuming. For LP solving, the Gurobi algorithm has polynomial run-time in the number of variables.

However, we do have a huge number of variables. Empirically, we need to cope with about $\#constraints \times \#statements^{1.2}$ variables.

**Anecdotal examples.** Table 8 gives a few anecdotal outputs with scores returned by DICE. Note that the scores produced do not represent probabilities, but dimension-wise ranks (i.e., we percentile-normalized the scores produced by DICE, as they have no inherent semantics other than ranks). For instance, `be at shed` was found to be much more typical than `be at pet zoo` for `snake`, while salience was the other way around. Note also the low variation in ConceptNet scores, i.e., in addition to being unidimensional, this low variance makes any ranking difficult.

## Appendix C. Web demonstrator

The results of running DICE on ConceptNet and Quasimodo are showcased in an interactive web-based demo. The interface shows original scores from these CSK collections as well as the per-dimension scores computed by DICE. Users can explore the values of individual cues, the priors, the taxonomic neighborhood of a subject, and the clauses generated by the rule grounding. The demo ready to be deployed online; for anonymous reviewing, we only give screenshots in Figure 1.

From a landing page (Fig. 1(a)), users can navigate to individual subjects like *band* (Fig. 1(b)). On pages for individual subjects, taxonomic parents and siblings are shown at the top, followed by commonsense statements from ConceptNet and Quasimodo. For each statement, its normalized score or percentile in its original CSK collection, along with scores and percentiles along the four dimensions as computed by DICE, are shown. Colors from green to red highlight to which quartile a percentile value belongs. On inspecting a specific statement, e.g., *band: hold concert* (Fig. 1(c)), one can see related statements used for computing basic scores, along with the values of the priors and evidence scores. Further down on the same page (Fig. 1(d)), the corresponding materialized clauses from the ILP, along with their weight $\omega^c$, are shown.

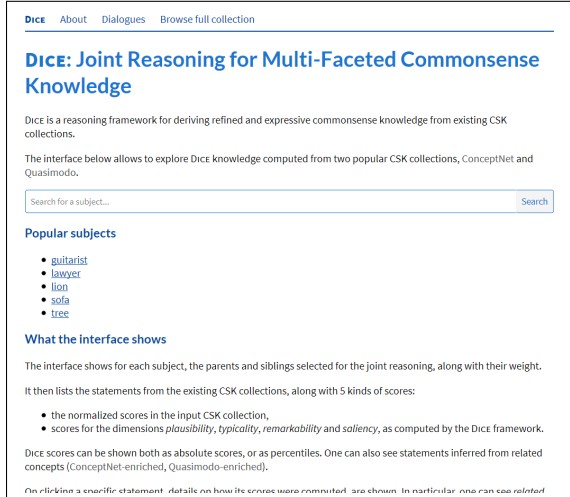

(a) Demo landing page.

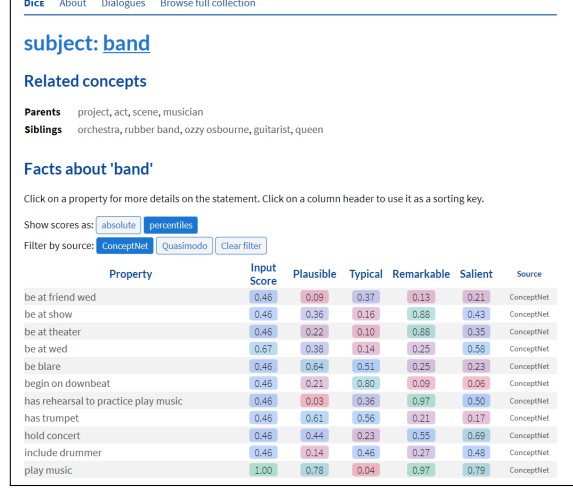

(b) List of statements for subject *band*.

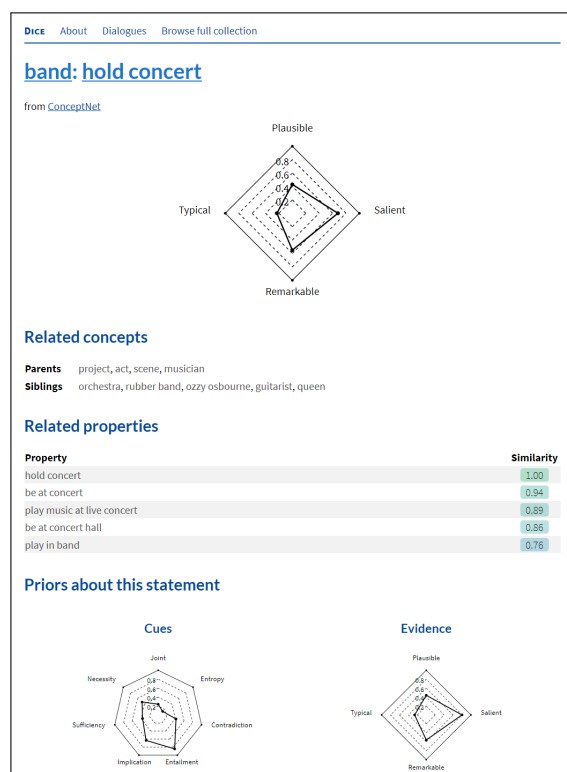

(c) Scores and neighbourhood for statement *band: hold concert*.

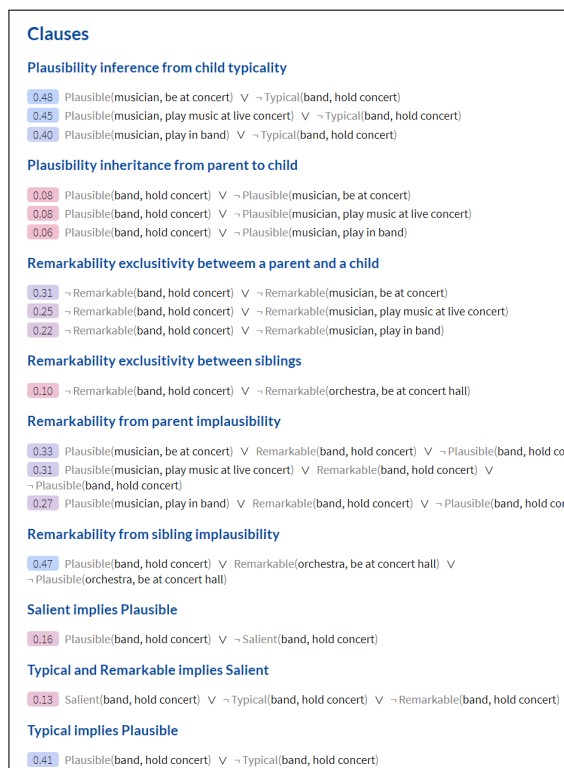

(d) Materialized clauses for statement *band: hold concert*.

Figure 1: Screenshots from the web-based demonstration platform.