# OpenReview forum: "Joint Reasoning for Multi-Faceted Commonsense Knowledge"
_AKBC.ws/2020/Conference — AKBC 2020_

### Official Review · AnonReviewer2 · 2020-03-24
**Interesting idea, but strange results and not well aware of prior work**

**Rating:** 4
**Confidence:** 5

**Review:**

Summary: The authors propose a “multi-facet” notion of scoring common sense knowledge, arguing that it is not sufficient to have just a single “confidence” score, but instead propose four distinct values: plausibility, typicality, remarkability, and salience.  These values are codified into a series of constraints that conform to their natural language definitions, leading to an approach for extending existing common sense KBs based on this new multi-faceted perspective.
Unfortunately after a detailed inspection at the resulting dataset, the relationship between the proposed scores seems surprising: e.g. in the majority of cases, plausibility score is lower than saliency and typicality, and the scores are highly correlated among each other. Moreover, there is little detail on how the extended versions of CSK datasets were constructed and filtered. Also, some of the design choices for calculation and aggregation of basic scores (Sections 5.1 and 5.2) are not convincing and should be explained.  Finally, the authors seem unaware of a significant amount of prior work which would need to be addressed to properly contextualize this contribution.

Review: The paper investigates how to associate useful scores to common-sense statements. CSK databases typically provide a single “confidence” score, which the authors claim is too limiting.  The authors primarily motivate their investigation from the downstream goal of dialogue: in such a setting, it may be that a given fact holds nearly all the time, e.g. that “people breathe”, but it isn’t very interesting in most circumstances and not worth discussing.  Other things may be interesting, but so unlikely that it would be strange to discuss it.  This general observation has been made previously, such as in:

Jonathan Gordon and Benjamin Van Durme. 2013. Reporting Bias and Knowledge Extraction. In Automated Knowledge Base Construction (AKBC) 2013: The 3rd Workshop on Knowledge Extraction, at CIKM

The authors go beyond this, proposing 4 distinct “facets” of common knowledge that each deserve a score.  I found this part of the paper interesting, the core idea worth pursuing.  However, this part of the paper would be significantly improved with more work in relating the proposal to significant prior work.  For example:

L.K. Schubert and M.H. Tong, "Extracting and evaluating general world knowledge from the Brown corpus", Proc. of the HLT/NAACL 2003 Workshop on Text Meaning, May 31, Edmonton, Alberta, Canada.

This article proposed the use of a 6-way categorical label on common sense, for purposes of human scoring.  It would be nice for the authors here to discuss this, and then argue for why their facets are better.

Related as well is the ordinal scale proposed in JOCI:

Sheng Zhang, Rachel Rudinger, Kevin Duh, and Benjamin Van Durme. 2017. Ordinal Common-sense Inference. Transactions of the Association for Computational Linguistics, 5:379–395.

Where plausibility, for example, was proposed as an ordinal value on a continuum.

This article by Schubert is the first, to my knowledge, that gave a coherent pitch on the idea of extracting common sense from a large corpus:

L.K. Schubert, "Can we derive general world knowledge from texts?", M. Marcus (ed.), Proc. of the 2nd Int. Conf. on Human Language Technology Research (HLT 2002), March 24-27, San Diego, CA, pp. 94-97.

 Especially salient here is the work on LORE:

https://cs.rochester.edu/research/lore/

Clark and Harrison released the DART collection:

Large-Scale Extraction and Use of Knowledge From Text. P. Clark, P. Harrison. Proc. Fifth Int Conf on Knowledge Capture (KCap) 2009

Maria Liakata and Stephen Pulman were another early example of a framework akin to Schubert's KNEXT:

Maria Liakata and Stephen Pulman. 2002. From Trees to Predicate Argument Structures. In Proceedings of COLING.

Moving to the experiments in this paper:, after looking at the score values on the ConceptNet part of the dataset, I find several things surprising. For instance, even though plausibility is a superset of typicality and saliency, in 56% and 77% cases plausibility score is lower than typicality and saliency, respectively, on the extended part of ConceptNet. It looks unexpected given the definition of these scores and inference rules 1--2. Additionally, even though the four proposed scores are designed to decompose the original confidence measure, there seems to be a lot of interdependence between them (e.g. the correlation between plausibility and saliency is over 88% on the extended part of ConceptNet).  I understand that this has a positive side effect, which is the ability to come up with a lot of inference rules between the four scores, but it should be stated more clearly whether that was the original design goal.

Given that the extended CSK datasets are a major contribution of this paper, it is also surprising to see so little detail about the actual extension process, both in the paper and in the appendix. For instance, at the end of the paper the authors mention that “inferred statements ... expand the original CSK data by about 50%”. It is not further explained how the entirety of the inferred statements is constructed. Given the abundance of inference rules (1--14), there can be a lot of inferred statements, so we assume some filtering was used to prune noisy statements for the final dataset, but we couldn’t find any details on filtering in the paper or appendix. Additionally, it is not clear what evaluation measure is used in Table 6.

The description of the proposed method is also not very convincing for the following reasons:
a. One of the important relationships between statements is the similarity, and your model relies on it in two places -- similar properties expansion and basic scores. The similarity between two strings is measured by the cosine similarity of weighted word embeddings. But word embeddings, especially for word2vec, are not reliable in this situation. E.g., I briefly looked into the database you released, the model gives two words “inhabit” and “live” with very different plausibility scores.
b. The features for the basic score are all heuristics. Why do you set an arbitrary threshold for quasi-probability, instead of using a soft weighted average over all statements? What's the semantics behind P[s] and P[p]? To validate the effectiveness of every feature, an ablation study should be done.
c. For textual entailment score, more details are needed. For the example you provide in the paper, how would the first sentence "Simba is a lion" be constructed given a CSK statement? Do you use a universal template to generate sentences for all relationships, or have specific templates for every relationship? I understand the second statement “Simba lives in  a pride” could be constructed by simple concatenation, but it’s not always grammatically correct, so how do you solve this problem?


I conclude the paper approaches an interesting problem and makes algorithmic and resource suggestions, but more should be done to ground the contributions in prior work, the current results are underwhelming, and there are several un-/under-explained design decisions which need to be addressed before publication.

The following are just some of the surprising examples found in basic digging through through the resource, the sort that I would anticipate as part of writing the article with associated discussion:

(Original ConceptNet with Dice scores)
“Person attempt suicide” has (plausibility 0.94, typicality 0.96, remarkable 0.007)
“Person iron pant” has (plausibility 0.95, typical 0.96, remarkable 0.0006)

Looking at the most remarkable facts in Extended ConceptNet (sorting on that value) shows a lot of odd knowledge.

Looking at the most salient info about Italy, or China, also shows strange facts.

Etc.

---

> ### Author Response · Authors · 2020-04-09
> **Revised paper**
>
> Thanks for the helpful comments. We revised parts of the paper; the changes are typeset in blue color.
>
> == Interpretation of scores ==
>
> This is a misunderstanding. The scores merely serve to rank statements by each facet; the numbers are not comparable across facets and were not meant to achieve this. Nevertheless, ranking per facet is very useful towards high-quality and wide-coverage CSK collections.
>
> The scores per facet are naturally correlated, partly because they reflect intertwined aspects of reality, but mostly because our soft constraints couple them. Nevertheless, the absolute numbers of scores do not necessarily have intuitive interpretations, and are still not comparable across facets.
>
> == Design choices for basic scores (priors) ==
>
> We do not claim that the prior scoring model is the best one. We simply made pragmatic choices. There are alternatives, which could be explored. Our emphasis and contribution is on the soft-constraint-based reasoning over multiple candidate statements, jointly for all facets and considering taxonomic neighborhoods. The reasoning part of our framework would easily work with alternative models for priors.
>
> == Prior works on CSK inference ==
>
> We appreciate the literature pointers, and apologize for missing out on this relevant work. We have included a new paragraph on Knowledge Modalities in Section 2, where we discuss the most important ones of these papers, giving them proper credit.
>
> Indeed, the design thinking of these papers is inspirational and related to our approach. One may think of our paper as an operationalization and extension of these early works.
> However, we do not see a prior method that would capture all four facets together. Also, some of these works operate on a very different data model, motivated by NLI tasks rather than building a comprehensive CSK collection. For example, the JOCI data consists of sentence pairs and annotates entailment-like properties. But these sentences, from various corpora, are not suited, and presumably never intended, to acquire a large commonsense KB organized by crisp subjects and aiming to systematically cover properties of subjects. In terms of methodology, this line of prior research is also rather different (again because it seems driven by different goals). We do not see how these methods can be applied for Web-scale CSK.
>
> == Experiments ==
>
> Plausibility lower than typicality or salience:
> Scores are meaningful only for ranking statements within each facet. They are unsuited for comparison across facets.
>
> Knowledge enrichment:
> We expanded a concept’s properties by considering its two-hop neighborhood in the taxonomy, generating noisy candidates. This was fed into the reasoning to derive a cleaner set of new properties.
> We clarified this in the revised text; see Appendix B.2 Potential for Enrichment.
>
> Metric for Table 6:
> These are the Likert-scale values. We clarified this in the revised text; see Appendix B.2.
>
> == Method description and justification ==
>
> Limitations of embedding-based similarity:
> Models like word2vec clearly have limitations. Our reasoning framework and the scoring prior can easily plug in alternative models of embeddings or other similarity metrics. Newer models like BERT might ameliorate the shortcoming of word2vec. However, we would not expect break-through improvements this way, as we need to cope with very short snippets, reporting bias on our input sources, and the difficulties of BERT-like models to properly detect negation.
>
> Threshold for quasi-probability:
> Thresholding is needed, because in embedding space, too many statements are within some proximity. We need to prune this space.
>
> Semantics of P[s] and P[p]:
> These are the prior probabilities of observing s as a subject of any statement, or p as the property of any statement in the CSK collection.
>
> Effectiveness of features:
> We limited ourselves to a small number of hyper-parameters, to avoid hinging on too much labeled training data for supervision. Introducing feature weights and varying them in ablation studies is a good suggestion. Note, though, that experiments in this direction would be computationally expensive and may run for weeks or months. Hence considering fine-grained feature weights has been out of scope so far.
>
> Text entailment score:
> We use a generic placeholder X for subjects, generating, for example, “X chase deer”.. Grammaticality may suffer, but this has a negligible impact on entailment scores.
>
> == Comment on example statements ==
>
> We believe our method and its results are a substantial step forward towards clean and wide-coverage CSK collections. We do not claim, though, that we have fully reached this goal. Our CSK data still contains many spurious and weird statements. The experimental evaluation showed, though, that we improve the quality and coverage of pre-existing large CSK collections by a significant margin.

---

> > ### Comment · AnonReviewer2 · 2020-04-16
> > **Why are scores not more comparable across facets?**
> >
> > You design an ILP relaxed to an LP, where for example: if "t => p" (typical => plausible), then one of the CNF is "\neg t V  p", and the corresponding ILP constraint is "p + 1 - t >= z", where z is 0 or 1 indicating the satisfaction of this logic.
> >
> >
> > Shouldn't scores therefore be more comparable across different facets, given that the rules over which you are performing inference have constraints with those facets mixed together?
> >
> > But backing off from comparability: while they may not be perfectly calibrated, it still intended that "higher is better", and it remains the case that many rules scoring high seem nonsensical or worse.
> >
> > If this paper is accepted then I would hope to see more qualitative analysis of the contents of the resultant KB, stressing that while the approach in this article is interesting and novel, there is still a long way to go.

---

> > > ### Author Response · Authors · 2020-04-17
> > > **Scores semantics and scope for further improvement**
> > >
> > > Our final scores are the reduced costs of the LP variables, not the truth assignments (which would be binary). Intuitively, the reduced costs reflect the "importance" of a variable being set to 1 with regard to the overall objective function. As the coupling of facets is asymmetric (see rule 6 - typical for child implies plausible for parents), this "importance" is still not comparable across facets.
> > >
> > > We realize that our results do have limitations. On the other hand, none of the previous premier-published one-dimensional CSK collections achieved a convincing and noise-free ranking ability, and our results improve on these prior works. We would further elaborate on the limitations in an accepted version.

---

### Official Review · AnonReviewer1 · 2020-03-27
**Interesting paper where motivation of setup and evaluation could be stronger**

**Rating:** 7
**Confidence:** 4

**Review:**

The paper describes a method for predicting different facets ('plausibility', 'typicality', 'remarkability', 'saliency') of validity for common-sense facts.
The proposed method is a two-step process
 (1) prior weights on the facets are regressed from features (e.g. basic statistics of facts)
 (2a) inference rules are given that connect different facets/facts (considering textual similarity)
 (2b) a weighted and relaxed ILP/LP is solved to obtain facet scores for all facts considering prior weights and inference rules

The paper proposes an interesting approach to an interesting type of problem, it reports a substantial amount of work, and it is very well written.

However, I am not convinced regarding the following two central points:

a) Motivation, Choice and Definition of facets

A motivation why exactly those 4 facets are chosen is lacking, and the motivation given for single factes (e.g. for generation of funny comments) should be stronger.
Moreover, I wonder whether the references given in 3.1 really motivate/clarify the differences between the facets , e.g., is it really the case that the property captured Tandon 2014, Mishra 2017 is plausibility (SOME instances) whereas in contrast Speer and Havasi measure Typicality (MOST instances)? (I would think all three aim at typicality).
It would also help to give some insight how exactly the different facets were described to the annotators, to get a feeling what is really measured in the end.

b) Evaluation

Pairwise ranking for 800 pairs of instantiated facts with facets are annotated.
70% are used for estimating weights, and 30% are used for evaluation.
Now the interesting question, that should be critically investigated, is whether the main part of the pipeline (step 2, the rule-based inference system) actually improves on the simple regression model (step 1).
As is reported in the paper, the regression alone model gives preference accuracy of 58%, whereas the full model gives 66%.
Putting everything together, this translates into a difference of 19 cases (0.3 * (0.66-0.58) * 800), and efforts to investigate the statistical significance of this difference (type of test, p-value etc) should be reported (but are not).


Other points:
 - there is a lot of important/central material in the appendix. Given that theoretically it should not be necessary that readers/reviewers read it, I wonder whether a 10 page conference paper is the right format for this work.

Small questions/comments:
 - "crisp" noun: do I understand it correctly that only single word concepts can be subjects? why?
 - appendix A.3: "partitions overlap" - then partition is not the right word. subset?

---

> ### Author Response · Authors · 2020-04-09
> **Revised paper**
>
> Thanks for the helpful comments. We revised parts of the paper; the changes are typeset in blue color.
>
>
> == Motivation for choice of facets ==
>
> The three main facets - plausibility, typicality and salience - have all been considered in prior works on Web-scale CSK, although, to the best of our knowledge, no project has considered all of them together. Remarkability is a new facet, motivated by our leveraging of taxonomic relations. Note, though, that even if we dropped remarkability as a facet, reasoning over taxonomic neighbors is still beneficial for accuracy and coverage. Moreover, no prior project has jointly tackled the three main facets at this scale.
>
> We revised the text on the design rationale, in Section 3.1. We also included a new paragraph on additional related works in Section 2.
>
>
> == Semantic focus of existing CSK collections ==
>
> TupleKB [Mishra 2017] explicitly mentions that their focus is plausibility (Section 3.3.4: “the semantics we apply to tuples (and which we explain to Turkers) is one of plausibility: If the fact is true for some of the arg1’s, then score it as true”).
>
> WebChild [Tandon 2014] does not explicitly mention plausibility in the paper, but we verified this choice in personal communication with the authors.
>
> ConceptNet does not explicitly mention its semantics in papers, but relation definitions refer to typicality (e.g., “/r/CapableOf: Something that A can typically do is B.” [https://github.com/commonsense/conceptnet5/wiki/Relations].
> User interfaces frequently contain the modifier “likely”, or imply it. Arguably, by asking humans, ConceptNet also aims for salience, but its coverage in this regard is very limited.
>
> KNEXT [Schubert 2002] (newly added to the Related Work section) primarily aimed at typicality. Recent work on ordinal CSK [Zhang 2017] addresses the spectrum from typicality to plausibility, but its data model is natural language sentences rather than concept-property or SPO statements. So it is not comparable to Web-scale structured CSK collections.
>
>
> == Guidelines for MTurk annotators ==
>
> We included a short section on the guidelines for the crowd workers in Appendix B.1.
>
>
> == Statistical significance ==
>
> Although, the absolute numbers in the evaluation set are not that large, the reported differences are statistically significant, by one-tailed t-tests.
> We added brief clarification on statistical tests, both in the paper and the appendix.
>
>
> == Material in main paper vs. appendix ==
>
> We admit that the splitting between paper and appendices is far from optimal. We tried our best to make the 10-page paper itself complete and coherent. The appendices serve as supplementary material for this submission. When the paper is published, all appendices can go to a web page on the entire project. This will comprise descriptions of implementation and experimental details (i.e., what is now in the appendices), the CSK collections, evaluation data, source code, and the interactive CSK browser (Appendix C).
>
>
> == Comment on “crisp nouns” ==
>
> We clarified this in the text, see beginning of Section 3 on page 4.
>
>
> == Comment on “overlapping partitions” ==
>
> We clarified this in the text, see Appendix A.3 on page 14

---

> > ### Comment · AnonReviewer1 · 2020-04-21
> > **Response after rebuttal**
> >
> > I would like to thank the authors for addressing my concerns and updating the paper.
> > In light of the improvements to the paper, I am changing my rating from 6 to 7.

---

### Official Review · AnonReviewer3 · 2020-03-29
**Models for a more nuanced view of commonsense knowledge, but with some open questions**

**Rating:** 8
**Confidence:** 5

**Review:**

Synopsis:
This paper attempts to classify different scoring mechanisms introduced in widely used KB resources into four facets:  plausibility, typicality, remarkability, and salience. To model these scores, the authors introduce logical dependencies between these scores and use an ILP-based reasoning model (enhanced by vector-space similarity models) to learn these scores. The additional nuance of these measures is an advance in commonsense knowledge and the underlying model provides a compact and convincing set of constraints behind these scores, but some of the methodological details are missing or could be improved to allow a stronger impact.


- Clarity: generally easy to read and understand, with some room for improved prose
- Quality: thoughtful approach to the problem, but methodological issues raise doubts
- Originality: integrates ideas from prior work in a new and novel way
- Significance: meaningful contribution to CSK literature

Most knowledge base projects report a score for each fact. However, the semantics of these scores can be difficult to ascertain. In this paper, the authors propose four differing semantics for assessing these facts and a model for estimating these scores. Although these semantics have been, to some extent, been proposed in prior work, this appears to be the first work that integrates all four concepts into a single model.  Unfortunately, the last facet score, salience, seems more subjective. Someone living in the African bush might have a very different view of what is salient about hyenas than an office worker, and as such separating the cultural background of the viewer from the score seems short-sighted.

The choice to combine properties and objects could be supported more strongly. In particular, whether such a method is appropriate for more sophisticated knowledge representation schemes where qualifiers are present on a property, or how this method would work in more open-world settings where properties or objects can consist of multiword phrases. The use of embeddings addresses that concern, but could still be emphasized.

The codification of scoring semantics is done through a set of logical rules relating the four score types to each other both for a single entity and between child and sibling entities in the commonsense knowledge base. These logical rules are used as input to an integer linear programming formulation, and although the problem is intractable, several approximations including partitioning the problem are used to allow tractable inference.

One surprising choice is introducing more general statements into the candidate set, by assuming parents have the same properties as their children. Is this necessary or a hack to make the method work? The same question could be asked about word-level embeddings used to score similar properties - what happens when these aren't used, won't IDF weighting always downweight the properties since most commonsense knowledge resources use a few properties, and why not use a more sophisticated language model like RoBERTa?

The choice of using ILPs rather than more flexible probabilistic frameworks that have scalable inference engines (ProbLog, PSL, ProPPR, etc.) is confusing, particularly it seems that substantial effort was required to get ILPs to work in this setting. The reliance on hyperparameters, specifically $\omega_r$ are not particularly clear, and it would be interesting to know if uniform rule weights diverge substantially from learned rule weights, especially since the reported ablation results suggest the priors dominate the constraints.

Further details of the MTurk experiment would be helpful, particularly since some ratings may be subjective. Were there limitations on the geographical location, educational background, or other demographic characteristics to participate? Where the workers compensated fairly? How were instances sampled from large commonsense corpora to ensure balanced coverage?

Some specific suggestions:
- Abbreviations (ca. incl.) should be avoided in scientific prose
- Illustrating the space of plausible, typical, remarkable, and salient facts would be helpful. A Venn diagram of four concentric circles would illustrate this clearly.
- Compare to more tractable modeling frameworks (e.g., probabilistic soft logic has been used for knowledge graph analysis and supports confidence values)?
- Specify where child and sibling relationships are coming from in the main draft
- Provide more details about the MTurk sampling since it seems as if there is a bias towards the DICE facts

Overall, I believe this paper provides a thoughtful look at scoring commonsense knowledge and would be welcomed by the AKBC community as a step forward in improving the approach to building and evaluating commonsense. Clearer explanation of some of the methodological choices, the inclusion of additional probabilistic models, and discussion of other knowledge representation paradigms would strengthen this paper.

---

> ### Author Response · Authors · 2020-04-09
> **Revised paper**
>
> Thanks for the helpful comments. We revised parts of the paper; the changes are typeset in blue color.
>
> == Subjectivity of salience ==
>
> We agree that the notion of salience cannot be perfectly objective, as it is human-centered. However, we believe that there is wide inter-subjective agreement within socio-cultural groups that are fairly large and reasonably homogeneous. The example of “someone living in the African bush” is too specific and misleading. We would rather aim at high agreement within groups like Central European Adults, American College Graduates or Fareast Asian Women. So far, we ignore this dimension of socio-cultural CSK and focus on the prevalent case of Western World Adults (i.e., the group that produces most English-languge contents on the Internet). We are not aware of any work along these lines of socio-cultural knowledge, but fully agree that this would be good direction to explore.
>
> == Motivation for combining predicates and objects ==
>
> We revised the text on the motivation, see second paragraph of Section 3 on page 4.
>
> == Surprising effects from taxonomic parent-child relations ==
>
> This is a misunderstanding. We assume that parent statements also apply to children, but in the converse direction only assume that *typical* children statements are *plausible* for their parents. We expanded the description of using taxonomic relations for knowledge enrichment; see Appendix B.2 Enrichment Potential.
>
> == Use of word-level embeddings ==
>
> We do not rely on taxonomic relations here to overcome sparseness. Our notion of properties concatenates the P and O arguments of pre-existing CSK collections such as ConceptNet, for example, considering embeddings for “is capable to carry a trunk”, “is capable to lift logs from the ground”, “is capable to lift a tree” etc. ConceptNet has only few predicates, but many P+O combinations for the same subject, such as “elephant”.
>
> == Using more advanced embeddings such as RoBERTa ==
>
> This is a potential direction for future work.
>
> == Hyper-parameters for rule weights ==
>
> The rules are still a heuristic model where different rules may have different influence. Therefore, we introduce hyper-parameter weights. Using our withheld data, the learned rule weights were [0.48, 0.28, 0.14, 0.09, 0.48, 0.66, 0.58, 0.51, 0.42, 0.22, 0.13, 0.14, 0.60, 0.85] for rules 1 through 14. We admit that we did not perform sensitivity experiments on hyper-parameter values, but we can easily re-run our entire pipeline to investigate the influence of hyper-parameter choices. This will take a couple of weeks, hence not addressed in the revision so far.
>
> == Choice of using ILP ==
>
> We did consider graphical models such as PSL, but still saw tractability issues as MAP inference alone is not sufficient for our purpose. Conversely, we found that the industrial-strength ILP solver from Gurobi works fairly well. With our partitioning heuristics, the resulting ILPs are solvable within acceptable time on commodity servers.
> We include a brief discussion of this point at the end of Section 4 on page 8.
>
> Using ILP and relaxed LP is fairly simple (with the proper technical background). This includes also the use of reduced costs, for someone familiar with mathematical optimization. These techniques have not been widely used in KB/CSK research, but they are standard in optimization. We do not claim novelty on these methods; our contribution is to judiciously leverage them for CSK acquisition.
>
> == Further details on MTurk evaluation ==
>
> We included a short section on the guidelines for the crowd workers in Appendix B.1. We also added clarification on statistical significance tests, both in the paper and the appendix.
>
> The samples shown to the annotators were randomly drawn proportionally to their scores (with higher likelihood of being sampled for statements with higher scores). We believe this is fair with regard to baselines.
>
> == Presentation issues ==
>
> We tried our best to clarify issues and enhance the presentation. However, the 10-page limit for the paper does not allow accommodating all suggestions.

---

### Decision · Program_Chairs · 2020-05-01

**Decision:**

Accept

**Comment:**

This paper presents a method for classifying scoring mechanisms in KB resources, applied to commonsense knowledge bases. The reviewers see the application to common-sense knowledge bases as a strong point of the paper and agree that the goal is worthwhile, and that the paper is well-written. However, the reviewers had reservations about the grounding in related work, which should be improved, and maintain that the design decisions should be explained better.